# Local policy search with Bayesian optimization

**Sarah Müller**[*1,4]   **Alexander von Rohr**[*1,2,3]   **Sebastian Trimpe** [1,2]

[1]Max Planck Institute for Intelligent Systems, Stuttgart, Germany
[2]Institute for Data Science in Mechanical Engineering, RWTH Aachen University, Germany
[3]IAV GmbH, Gifhorn, Germany
[4] Institute for Ophthalmic Research, University of Tübingen, Tübingen, Germany
`sar.mueller@uni-tuebingen.de`
`{vonrohr, trimpe}@dsme.rwth-aachen.de`

## Abstract

Reinforcement learning (RL) aims to find an optimal policy by interaction with an environment. Consequently, learning complex behavior requires a vast number of samples, which can be prohibitive in practice. Nevertheless, instead of systematically reasoning and actively choosing informative samples, policy gradients for local search are often obtained from random perturbations. These random samples yield high variance estimates and hence are sub-optimal in terms of sample complexity. Actively selecting informative samples is at the core of Bayesian optimization, which constructs a probabilistic surrogate of the objective from past samples to reason about informative subsequent ones. In this paper, we propose to join both worlds. We develop an algorithm utilizing a probabilistic model of the objective function and its gradient. Based on the model, the algorithm decides where to query a noisy zeroth-order oracle to improve the gradient estimates. The resulting algorithm is a novel type of policy search method, which we compare to existing black-box algorithms. The comparison reveals improved sample complexity and reduced variance in extensive empirical evaluations on synthetic objectives. Further, we highlight the benefits of active sampling on popular RL benchmarks.

## 1   Introduction

Reinforcement learning (RL) is a notoriously data-hungry machine learning problem, where state-of-the-art methods easily require tens of thousands of data points to learn a given task [1]. For every data point, the agent has to carry out potentially complex interactions with its environment, either in simulation or in the physical world. This expensive data collection motivates the development of sample-efficient algorithms. Herein, we consider policy search problems, a type of RL technique where we directly optimize the parameters of a policy with respect to the cumulative reward over a finite episode. The collected data is utilized to estimate the direction of local policy improvement, enabling the use of powerful optimization techniques such as stochastic gradient descent. Policy gradient methods (e.g., [2–5]) usually rely on random perturbations for data generation, e.g., in the form of exploration noise in the action space or stochastic policies, and do not reason about uncertainty in their gradient estimation. However, innate in the RL setting is the ability to actively generate data, allowing the agent to decide on *informative queries*, thereby potentially reducing the amount of data needed to find a (local) optimum. Active sampling has the potential to allow those algorithms to improve sample complexity, reducing the number of environment interactions.

---

[*]Equal contribution

35th Conference on Neural Information Processing Systems (NeurIPS 2021).

In contrast to random sampling, Bayesian optimization (BO) [6] is a paradigm to optimize expensive-to-evaluate and noisy functions in a sample-efficient manner. At the core of BO is the question of how to query the objective function efficiently to maximize the information contained in each sample. By building a probabilistic model of the objective using past data and, critically, *prior knowledge*, the algorithm can reason about how to query a noisy oracle to solve the optimization task. Since RL can be framed as a black-box optimization problem, we can use BO to learn policies in a sample-efficient way. However, even though BO has been used to tackle RL, these approaches are often restricted to low-dimensional problems. One reason is that BO aims to find a *global* optimum; hence, without further assumptions, BO algorithms need to model and search the entire domain, which needs a lot of data and gets exponentially more

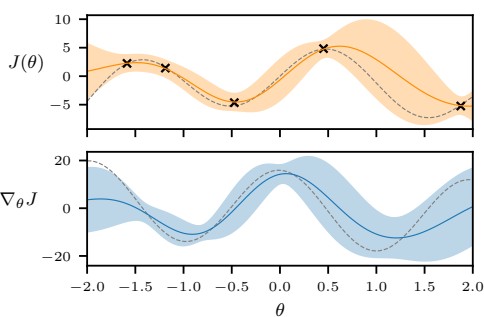

Figure 1: Estimation of a Jacobian GP model (bottom) of a 1-dimensional objective function (top). The model has function observations (black crosses), but is able to form a posterior belief over the gradient. Uncertainty for the Jacobian model is reduced *between* samples. An active sample strategy can improve gradient estimates.

difficult as the dimensionality increases. Additionally, as the amount of data grows so does the computational complexity of probabilistic models, which becomes a significant problem. However, the success of RL algorithms using policy gradient methods indicates that for many problems it is sufficient to find a locally optimal policy.

Our proposed algorithm combines the strength of gradient-based policy optimization with active sampling of policy parameters using BO. We thereby improve the computational complexity of BO methods on the one hand, and the sample-inefficiency of gradient-based methods on the other hand, especially when proper prior knowledge is available. We achieve these improvements by explicitly learning a probabilistic model of the objective in the form of a Gaussian process (GP) [7]. From this model, we can jointly infer the objective and its gradients with a tractable probabilistic posterior. The resulting Jacobian estimate includes all data points, rendering data usage more efficient. Further, the algorithm infers informative queries from the uncertainty of the probabilistic model to improve the estimate of the local gradient. While in this paper we adapt the setting of Mania et al. [1] and assume access to zeroth-order information only, the algorithm extends straightforwardly to policy gradient algorithms where additional first-order information is available. In summary, the contribution of this paper is a local BO-like optimizer called *Gradient Information with BO* (GIBO). The queries of GIBO are chosen optimally to minimize uncertainty about the Jacobian. GIBO uses a local GP model for active sampling and gradient estimation and can be used with existing policy search algorithms. Using only zeroth order information, GIBO is able to

- significantly improves sample complexity in extensive within-model comparisons, i.e., when accurate prior knowledge is available;
- is able to solve RL benchmark problems in a sample efficient manner; and
- reduces variance in the rewards when compared to non-active sampling baselines.

## 2 Preliminaries

This work presents a local optimizer with active sampling. The objective function and its derivative's joint distribution are modeled using a GP. Since we have developed the optimizer with the RL application in mind, we also introduce the RL problem. For the sake of brevity, we refer the reader to [7] and [8] for an introduction into GPs and BO, respectively.

### 2.1 Problem setting

In the following we phrase policy search as a black-box optimization problem. For a parameterized policy $\pi_\theta : \Theta \times \mathcal{S} \to \mathcal{A}$ that maps states $s \in \mathcal{S}$ and the static policy parameters $\theta \in \Theta$ to actions $a \in \mathcal{A}$, we use the same performance measure as in policy gradient methods for the episodic case.

Hence, the objective function $J : \mathbb{R}^d \to \mathbb{R}$ is defined as

$$J(\theta) = \mathbb{E}_{\pi_\theta} \left[ \sum_{i=0}^{I} r_i \right],$$

where $\mathbb{E}_{\pi_\theta}$ is the expectation under policy $\pi_\theta$, $r_i$ is the reward at time step $i$, and $I$ the length of the episode. A BO query is equivalent to the return of one rollout following the policy $\pi_\theta$ in the environment. The expected episodic reward is entirely determined by choice of policy parameters (and the initial conditions). Thus, the optimizer explores the reward function in the parameter space rather than in the action space. Since initial conditions might vary and the environment can be non-deterministic, reward evaluations are noisy.

Policy search herein is abstracted as a zeroth-order optimization problem of the form

$$\theta^* = \arg\max_{\theta \in \Theta} J(\theta), \tag{1}$$

where $\theta$ is the variable and $\Theta \subset \mathbb{R}^d$ a bounded set. To solve (1), an optimization algorithm can query an oracle for a noisy function evaluation $y = J(\theta) + \omega$. We assume an i.i.d. noise variable $\omega \in \mathbb{R}$ to follow a normal distribution $\omega \sim \mathcal{N}(0, \sigma^2)$ with variance $\sigma^2$. We do not assume access to gradient information or other higher-order oracles for conciseness. Albeit, GIBO requires that the following critical assumption is fulfilled:

**Assumption 1.** *The objective function $J$ is a sample from a known GP prior $J \sim GP(m(\theta), k(\theta, \theta'))$, where the mean function is at least once differentiable and the covariance function $k$ is at least twice differentiable, w.r.t. $\theta$.*

This is the standard setting for BO with the addition that the mean and kernel need to be differentiable, which is satisfied by some of the most common kernels such as the squared exponential (SE) kernel. In the empirical section, we investigate the performance of the developed algorithm with and without Assumption 1 holding true.

## 2.2 Jacobian GP model

Since GPs are closed under linear operations, the derivative of a GP is again a GP [7]. This enables us to derive an analytical distribution for the objective's Jacobian, which we can use as a proxy for gradient estimates and enable gradient-based optimization.

Following Rasmussen and Williams [7], the joint distribution between a GP and its derivative at the point $\theta_*$ is

$$\begin{bmatrix} \bar{y} \\ \nabla_{\theta_*} J_* \end{bmatrix} \sim \mathcal{N} \left( \begin{bmatrix} m(X) \\ \nabla_{\theta_*} m(\theta_*) \end{bmatrix}, \begin{bmatrix} K(X, X) + \sigma^2 I & \nabla_{\theta_*} K(X, \theta_*) \\ \nabla_{\theta_*} K(\theta_*, X) & \nabla_{\theta_*}^2 K(\theta_*, \theta_*) \end{bmatrix} \right), \tag{2}$$

where $\bar{y}$ are the $n$ zeroth-order observations, $X \subset \Theta$ are the locations of these observations $X = [\theta_1, \ldots, \theta_n]$, and $K$ the covariance matrix given by the kernel function $k : \Theta \times \Theta \to \mathbb{R}$. The posterior can be derived by conditioning the joint Gaussian prior distribution on the observation [7]

$$p\left(\nabla_{\theta_*} J_* \big| \theta_*, X, \bar{y}\right) \sim \mathcal{N}(\mu_*', \Sigma_*')$$

$$\mu_*' = \underbrace{\nabla_{\theta_*} m(\theta_*)}_{\in \mathbb{R}^d} + \underbrace{\nabla_{\theta_*} K(\theta_*, X)}_{\in \mathbb{R}^{d \times n}} \underbrace{\left(K(X, X) + \sigma^2 I\right)^{-1}}_{\in \mathbb{R}^{n \times n}} \underbrace{(\bar{y} - m(X))}_{\in \mathbb{R}^n} \in \mathbb{R}^d \tag{3}$$

$$\Sigma_*' = \underbrace{\nabla_{\theta_*}^2 K(\theta_*, \theta_*)}_{\in \mathbb{R}^{d \times d}} - \underbrace{\nabla_{\theta_*} K(\theta_*, X)}_{\in \mathbb{R}^{d \times n}} \underbrace{\left(K(X, X) + \sigma^2 I\right)^{-1}}_{\in \mathbb{R}^{n \times n}} \underbrace{\nabla_{\theta_*} K(X, \theta_*)}_{\in \mathbb{R}^{n \times d}} \in \mathbb{R}^{d \times d}.$$

**Remark 1.** *Note that the term $(K(X, X) + \sigma^2 I)^{-1}$ with the highest computational cost ($\mathcal{O}(N^3)$) is the same term that is used to compute the posterior over $J$. Therefore, calculating the Jacobian does not add to the computational complexity once a GP posterior has been computed.*

Any twice differentiable kernel is sufficient for the presented framework, but we assume a SE kernel for the remainder of the paper. For the derivatives of the SE kernel function see Appendix A.1. For a visual example of function- and the Jacobian-posterior, refer to Fig. 1. The figure indicates that

a zeroth-order oracle is enough to form a reasonable belief over the function's gradient. Moreover, Fig. 1 shows that the uncertainty about the Jacobian gets reduced *between* query points more so than at the query points themselves. To minimize uncertainty about the Jacobian at a specific point, it intuitively makes sense to space out query points in its immediate surrounding. Herein, we formalize this intuition and formulate an optimization problem that sequentially decides on query points that provide the most information about the Jacobian.

## 2.3 Related work

In the presented contribution we focus on the benefits of active sampling in policy search, specifically on sample efficiency. Therefore, this section focuses on active sampling in model-free RL setting using probabilistic uncertainty estimations. Most of the literature in this setting is based on BO, but generating informative samples is also discussed in literature regarding evolutionary strategies as well as in policy gradient methods.

Bayesian optimization as an active sampling method has been used for global policy search, mostly in lower dimensional parameter spaces from 2–15 dimensions [9–13]. Global BO for RL, exemplified by the mentioned literature and without additional assumptions, is limited to relatively low dimensional problems for two reasons: (i) the computational complexity of global probabilistic models does not scale well with the number of data points, (ii) global optimization of high-dimensional non-convex objectives is a challenging problem to solve in general. To combat these problems local variants of BO have been proposed and applied to RL problems, see e.g., [14–17]. These works rely on restricting the search space of BO by a probabilistic belief over the optimums location [14] using rectangular trust-regions [15], learning a partition [16], or by staying close (as defined by the GP kernel) to past samples [17]. Restrictions in the parameter space avoid 'over-exploration' of high-dimensional search spaces and thereby encourage exploitation of (local) minima. Our proposed method delegates the exploitation to a gradient-based optimizer after exploring a local property, the function's derivative at the current iterate, for which a local search (and model) is sufficient. McLeod et al. [18] and more recently Shekhar and Javidi [19] suggest switching from global BO to a local gradient-based method once a locally convex region containing a low-regret solution has been identified, thereby improving convergence properties of BO. In [18] GIBO can replace the local optimizer of choice and in Shekhar and Javidi [19] GIBO can be used for optimal uncertainty reduction in the gradient estimate.

In general, a GP posterior can incorporate gradient information if the kernel is differentiable and a first-order oracle is available. Bayesian optimization methods that utilize gradient observations are known as first-order BO, and different approaches on how to include the derivative information in the model and acquisition functions have been proposed [20–23]. Since computing the joint posterior using first- and zeroth-order information is computationally expensive, Ahmed et al. [21] and Wu et al. [22] are using a single directional derivative instead of all partial derivatives. A first-order BO approach for RL, where the gradient information is actively used to decide on the following query, is introduced by Prabuchandran et al. [23]. The method therein actively searches for local optima by querying points where the gradient is expected to be zero. In contrast to this approach, we actively reduce local uncertainty of the Jacobian model and afterwards a gradient-based optimizer decides on the next location.

Reinforcement learning problems in the form of (1) can also be used by evolutionary methods such as [1, 24–26] and recently by policy gradient methods Faccio et al. [27]. These methods typically explore through random perturbations in the parameter space of the policy instead of active sampling. However, generating more informative samples improves evolutionary strategies. Maheswaranathan et al. [25] shows this by adapting the sampling distribution using surrogate gradient information such as previous estimates, and Choromanski et al. [26] uses determinantal point processes for informative samples.

Policies that generate more informative samples have helped to improve model-free RL algorithms' performance during the past decade; we mention three examples here: Levine and Koltun [28] propose so-called guiding samples in high reward areas using differential dynamic programming and model knowledge. Soft actor-critic (SAC) methods [3] add the policy's entropy to the reward function to encourage exploration and improve the variance of gradient estimates. Based on SAC an optimistic actor-critic algorithm is introduced in [29] with a different exploration strategy that samples more informative actions. To reduce variance in the gradient estimate, it is possible to use GIBO as a layer between the policy gradient estimator such as SAC and a gradient-based optimizer, e.g., stochastic

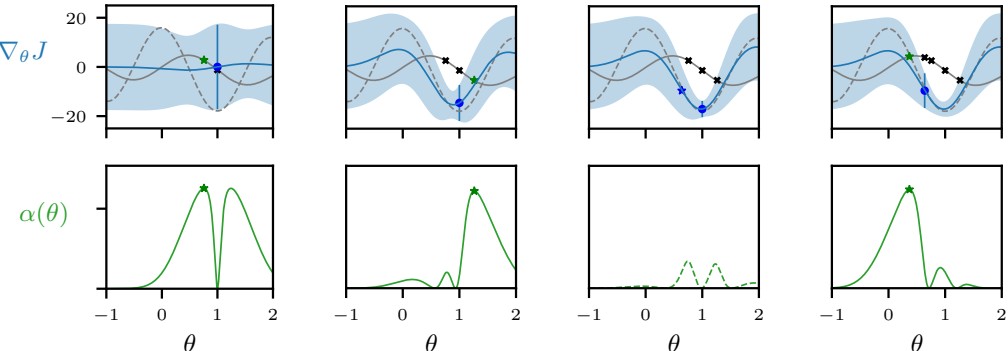

Figure 2: We visualize GIBO's active sampling process with a simple 1-dimensional function. The blue filled circle refers to the current parameter $\theta_t$. The figure shows 4 steps of the algorithm, where in the first two steps, the acquisition function $\alpha$ (solid green line) proposes two new query points (green stars) of the objective function $J$ (solid light grey line). With the history of sampling points (black crosses), the model of the Jacobian $\nabla_\theta J$ (in blue with confidence intervals) is updated, reducing uncertainty around the analytic Jacobian (dashed light grey line). The next step show a gradient ascent update step to $\theta_{t+1}$ (blue star) and the last step is again a suggested query after the update.

gradient ascent or Adam [30]. In future work GIBO can be extended to utilize state-of-the art policy gradient methods as an additional oracle for first-order information and help these methods reducing the variance of their gradient estimates through active sampling. Based on the posterior conditioned on all collected rewards, our algorithm can supply posterior gradient estimates and subsequent queries to evaluate.

To demonstrate the benefits of GIBO in a simple setup, we adopt the setting proposed by Mania et al. [1] as a baseline. Augmented Random Search (ARS) [1] assumes a black-box setting without access to gradient samples and estimates the gradient from the finite-difference of random perturbations, effectively solving RL problems. We replace the random sampling strategy of ARS with active sampling and the gradient estimation with a GP model. These changes improve the sample complexity and variance of ARS, especially when prior knowledge about the objective function is available.

## 3   Gradient informative Bayesian optimization

In this section, we introduce the proposed method GIBO. First, we define an acquisition function to reduce uncertainty for the Jacobian. Second, we outline the basic GIBO algorithm, including implementation choices.

### 3.1   Maximizing gradient information

We employ the BO framework to design a set of iterative queries maximizing gradient information. To this extend, we propose a novel acquisition function *Gradient Information (GI)* actively suggesting query points most informative for the gradient at the current parameters $\theta_t$. Acquisition functions measure the expected utility of a sample point based on a surrogate model conditioned on the observed data. The utility $U : \mathbb{R}^d \to \mathbb{R}$ of our method depends on a Jacobian GP model, the objective's observation data $\mathcal{D}$, and the current parameter $\theta_t$. It measures the decrease in the derivative's variance at $\theta_t$ when observing a new point $\theta$ of the objective function. Hence, we define the utility as the expected difference between the Jacobian's variance $\Sigma'(\theta_t|\mathcal{D})$ *before* and the Jacobian's variance $\Sigma'(\theta_t|\{\mathcal{D}, (\theta, y)\})$ *after* observing a new point $(\theta, y)$

$$\alpha_{\mathrm{GI}}(\theta|\theta_t, \mathcal{D}) = \mathbb{E}\left[U(\theta|\theta_t, \mathcal{D})\right] = \mathbb{E}\left[\mathrm{Tr}\left(\Sigma'(\theta_t|\mathcal{D})\right) - \mathrm{Tr}\left(\Sigma'\left(\theta_t|\{\mathcal{D}, (\theta, y)\}\right)\right)\right], \qquad (4)$$

where $\mathrm{Tr}$ denotes the trace operator and $\Sigma'(\theta_t|\mathcal{D})$ is the variance of the Jacobian's GP model evaluated at $\theta_t$

$$\nabla_\theta J\big|_{\theta=\theta_t} \sim \mathcal{GP}\left(\mu'(\theta_t|\mathcal{D}), \Sigma'(\theta_t|\mathcal{D})\right). \qquad (5)$$

The Jacobian's variance $\Sigma'(\theta_t|\{\mathcal{D}, (\theta, y)\})$ depends on the extended dataset $\{\mathcal{D}, (\theta, y)\}$. A property of the Gaussian distribution is, that the covariance function is independent of the observed targets $y$ as shown in Equation (3). Hence, we simplify the optimization over the expectation (see Appendix A.2) to

$$\arg\max_{\theta} \alpha_{\mathrm{GI}}(\theta|\theta_t, \mathcal{D}) = \arg\min_{\theta} \mathrm{Tr}\left(\Sigma'\left(\theta_t| [X, \theta]\right)\right), \tag{6}$$

where the variance only depends on a virtual data set $\hat{X} = [\theta_1, \ldots, \theta_n, \theta] =: [X, \theta]$. In conclusion, the most informative new parameter $\theta$ to query is only dependent on *where* we sample next and is independent of its outcome $f(\theta) = y$.

When we replace the Jacobian's variance in (6) with (3) and leave out constant factors we get

$$\theta^* = \arg\max_{\theta} \mathrm{Tr}\left(\nabla_{\theta_t} K(\theta_t, \hat{X}) \left(K(\hat{X}, \hat{X}) + \sigma_n^2 I\right)^{-1} \left(\nabla_{\theta_t} K(\theta_t, \hat{X})\right)^T\right). \tag{7}$$

Since the acquisition function only depends on the virtual data set, its optimization can be handled computationally efficient by performing the matrix inversion in (7) with Cholesky factor updates. Furthermore, since the Jacobian is a local property we can optimize (7) effectively using the of-the-shelf optimizer supplied by BoTorch [31] (L-BFGS-B) using multiple restarts.

## 3.2 The GIBO algorithm

The guided sequential search of the acquisition function for gradient estimates divides the resulting algorithm into two loops: An outer loop for iterative parameter updates and an inner loop where the acquisition function queries points to increase gradient information. The basic algorithm is given in Alg. 1.

---

**Algorithm 1** GIBO

1: **Hyperparameters**: stepsize $\eta$, hyperpriors for GP hyperparameters, number of iterations $N$ and number of samples for a gradient estimate $M$.
2: **Initialize**: place a GP prior on $J(\theta)$, set $\theta_0$ and $\mathcal{D} = \{\}$.
3: **for** $t = 0, \ldots, N$ **do**                                                       ▷ Parameter updates.
4:     Sample noisy objective function: $y_t = J(\theta_t) + \epsilon_t$
5:     Extend data set: $\mathcal{D} \leftarrow \{\mathcal{D}, (\theta_t, y_t)\}$
6:     GP hyperparameter optimization.
7:     **for** $m = 1, 2, \ldots, M$ **do**                                              ▷ Sample points for a gradient estimate.
8:         Get query point: $\hat{\theta} = \arg\max_{\hat{\theta}} \alpha_{\mathrm{GI}}(\hat{\theta}|\theta_t, \mathcal{D})$.
9:         Sample noisy objective function: $\hat{y} = J(\hat{\theta}) + \omega$.     ▷ Optionally: Use a policy gradient method
                                                                                          for additional derivative observations.
10:        Extend data set: $\mathcal{D} \leftarrow \{\mathcal{D}, (\hat{\theta}, \hat{y})\}$.
11:        Update the posterior probability distribution of $\nabla_\theta J$.
12:    **end for**
13:    $\theta_{t+1} = \theta_t + \eta \cdot \mathbb{E}\left[\nabla_\theta J\big|_{\theta=\theta_t}\right]$        ▷ Gradient ascent, or any other gradient based optimizer.
14: **end for**

---

## 3.3 Implementation choices

In the following, we introduce some details of our implementation of Algorithm 1 that further improve the performance and computational efficiency of our method.

**Local GP model.** Sparse approximation of GPs can be applied on BO when the computational burden of exact inference is too big [32]. In our case, however, we are only interested in estimating the local Jacobian at the current parameter $\theta_t$. We define a sparse approximation of the posterior at the current parameter $\theta_t$ heuristically with the last $N_m$ sampled points. Estimating a local model has the additional benefit of making the model selection and hyperparameter optimization simpler. We can approximate non-stationary processes locally by dynamically adapting hyperparameters.

**Local optimization of GI.** Following similar reasoning as above, we do not have to optimize the GI acquisition function globally since we expect informative points to be relatively close to the current parameter $\theta_t$ when using a SE kernel. Hence, we define our search bounds locally as $[\theta_t - \delta_b, \theta_t + \delta_b]$.

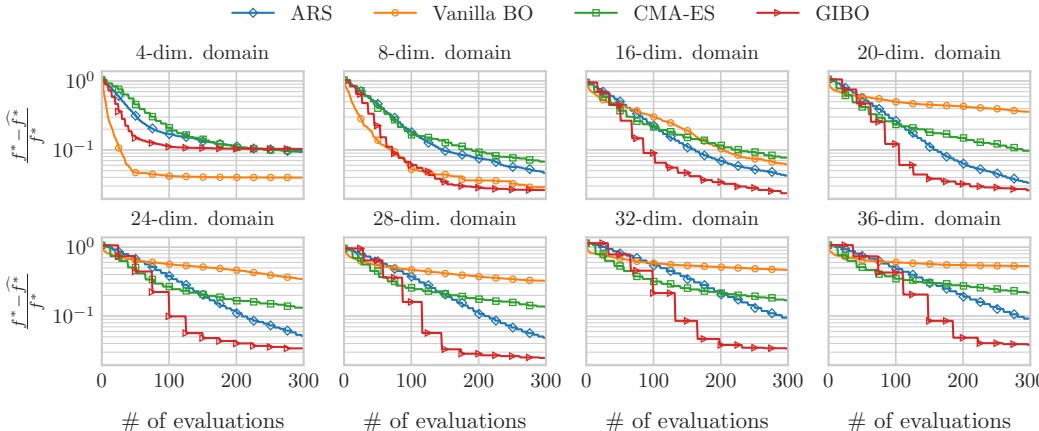

Figure 3: Within-model comparison: Mean of the normalized distance of the function value at optimizers' best guesses from the true global maximum for eight different dimensional function domains. 40 runs. Logarithmic scale.

**Gradient normalization.** The gradient is normalized with the Mahalanobis norm using the lengthscales of the SE kernel. Hence, the stepsize $\eta$ is adapted automatically to scale with the correlation between points. For the details see Appendix A.3.

**State normalization.** In the RL setting we can apply state normalization before we evaluate the policy to determine the next action. This has the same effect as data whitening for regression tasks and is beneficial when performing GP regression in unknown policy spaces. In case of a linear policy $\pi_\theta : \mathbb{R}^p \to \mathbb{R}^m$, $\pi_\theta(s) = As + b$ with bias $b \in \mathbb{R}^m$, states $s \in \mathbb{R}^p$, means of states $\mu_s \in \mathbb{R}^p$ and variances of states $\sigma_s \in \mathbb{R}^p$, state normalization can be defined by $\pi_\theta \left( \frac{s - \mu_s}{\sigma_s} \right) = A \left( \frac{s - \mu_s}{\sigma_s} \right) + b = A \cdot \frac{1}{\sigma_s} s - A \cdot \frac{\mu_s}{\sigma_s} + b$. State normalization is implemented in an efficient way that does not require the storage of all states. Also, we only keep track of the diagonal of the state's covariance matrix with Welford's online algorithm [33].

## 4 Empirical results

We empirically evaluate the performance of GIBO in three types of experiments. In the first experiment, we compare our algorithm on several functions sampled from a GP prior so that Assumption 1 is satisfied. In these *within-model comparisons* [34], we can show that GIBO outperforms the benchmark methods in terms of sample complexity and variance of regret, especially in higher dimension. In a second experiment, we perform policy search for a linear quadratic regulator (LQR) problem proposed by Mania et al. [1]. Finally, for RL environments of Gym [35] and MuJoCo [36], we show that GIBO reaches acceptable rewards thresholds faster and with significantly less variance than ARS. All data and source code necessary to reproduce the results are published at `https://github.com/sarmueller/gibo`.

### 4.1 Within-model comparison

We evaluate GIBOs performance as a general black-box optimizer on functions that satisfy Assumption 1. A straightforward way to guarantee this is by sampling the objective from a known GP prior. This approach has been called within-model comparison by Hennig and Schuler [34] but has likewise been used in other BO literature (e.g., [37, 38]). To show that GIBO scales particularly well to higher-dimensional search spaces, we analyze synthetic benchmarks for up to 36 dimensions.

The experiment was carried out over a $d$-dimensional unit domain $I = [0, 1]^d$. For each domain, we generate 40 different test functions. For each function, 1000 values were jointly sampled from a GP prior with a SE kernel and unit signal variance. To cover the space evenly, we used a quasi-random

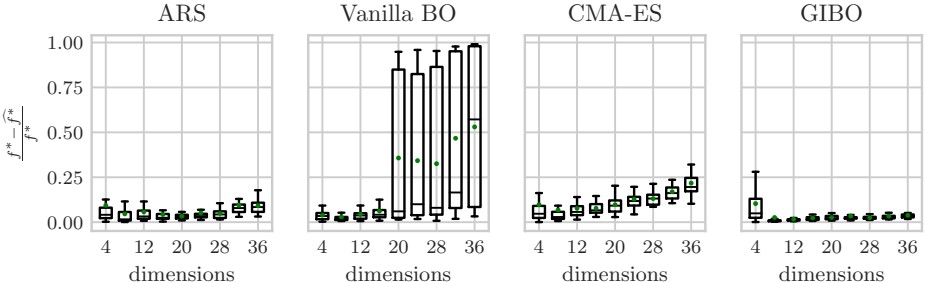

Figure 4: Within-model comparison: Boxplots (40 runs) show the normalized distance of optimizers' best found values after 300 function evaluations from the true global maximum. The whiskers lengths are 1.5 of the interquartile range; the black horizontal lines represent medians, green dots the means.

Sobol sampler. To perform experiments with comparable difficulty across different dimensional domains, we increase the lengthscales in higher dimensions by sampling them from the distribution $\ell(d)$, introduced in Appendix A.4. The resulting posterior means were the objective function. All algorithms were started in the middle of the domain $x_0 = [0.5]^d$ and had a limited budget of 300 noised function evaluations. The noise was Gaussian distributed with standard deviation $\sigma = 0.1$. A more detailed description of the experiments, including the true global maximum search and an out-of-model comparison, is given in Appendix A.4.

We compared our algorithm GIBO to ARS, CMA-ES [24] and standard BO with expected improvement [39] as acquisition function ('Vanilla BO'). To ensure a fair comparison, domain knowledge was passed to the ARS and CMA-ES algorithms by scaling the space-dependent hyperparameters with the mean of the lengthscale distribution $\ell(d)$. For details about the hyperparameters see Appendix A.7. The unknown hyperparameters were hand-tuned on a low dimensional example.

Fig. 3 shows the normalized difference between the global optimum and the function values of the optimizer's best guesses. The within-model comparison shows that our algorithm outperforms vanilla BO on all test functions, except for the 4-dimensional domain. With a limited budget of 300 function evaluations the proposed method, GIBO, achieved lower regret than the baseline methods, especially in higher dimensions. Further, GIBO was able to reduce the variance of obtained regret significantly, as shown in Figure 4, which indicates a consistently better performance.

## 4.2 Linear quadratic regulator

The classic LQR with known dynamics is a fundamental problem in control theory. In this setting, an agent seeks to control a linear dynamical system while minimizing a quadratic cost. With available

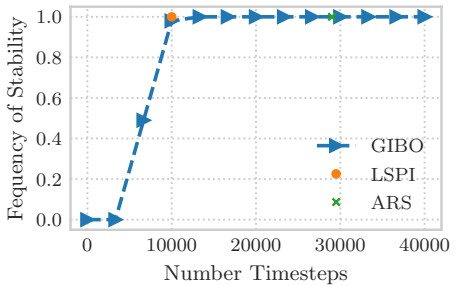
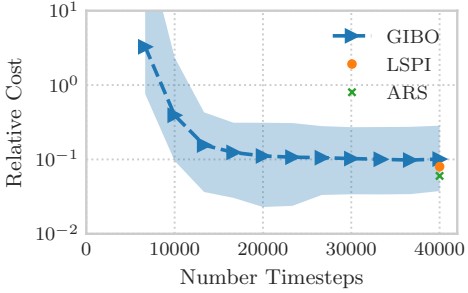

Figure 5: Results for the LQR experiment. Left: How frequently GIBO found stabilizing controllers in comparison to ARS and LSPI. The frequencies are estimated from 100 trials. Right: The sub-optimality gap of the controllers produced by GIBO compared to ARS and LSPI. The points along the dashed line denote the median cost, and the shaded region covers 2-nd to 98-th percentile out of 100 trials. Values for the benchmark methods in both images are estimated from [1].

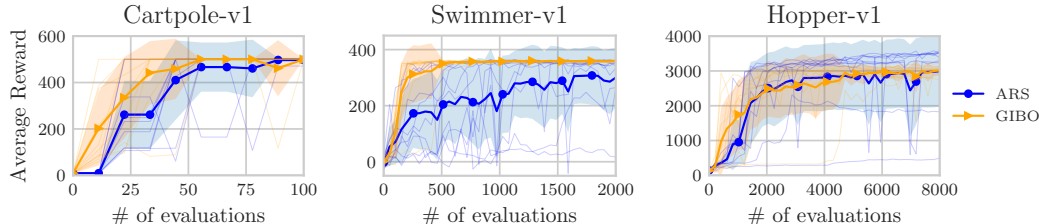

Figure 6: Training curves of GIBO and ARS for classic control and MuJoCo tasks, averaged over 10 trails (thin lines). The shaded regions show the standard deviation.

dynamics, the LQR problem has an efficiently determinable optimal solution. LQR with unknown dynamics, on the other hand, is less well understood. As argued in Mania et al. [1], this offers a new type of benchmark problems, where one can set up LQR problems with challenging dynamics, and compare model-free methods to known optimal costs. We compare GIBO against ARS and LSPI [40] on a challenging LQR instance with unknown dynamics, proposed by Dean et al. [41]. The reader is referred to Appendix A.6 for a complete introduction to the setup.

Fig. 5 shows the frequency of stable controllers found and the cost compared to the optimal cost for GIBO, ARS, and LSPI. On the left in Fig. 5 we observe that GIBO requires significantly fewer samples than ARS, equivalent to LSPI, to find a stabilizing controller. But we note that LSPI requires an initial controller $K_0$, which stabilizes a discounted version of the LQR problem. Neither GIBO nor ARS require any special initialization. All algorithms achieve similar regrets.

### 4.3 Gym and MuJoCo

Lastly, we evaluate the performance of GIBO on classic control and MuJoCo tasks included in the OpenAI Gym [35, 36]. The OpenAI Gym provides benchmark reward functions that we use to evaluate our policies' performance compared to policies trained by ARS. Mania et al. [1] showed that deterministic linear policies, $\pi_\theta : \mathbb{R}^p \to \mathbb{R}^m$, $\pi_\theta(s) = As + b$, are sufficiently expressive for MuJoCo locomotion tasks. Consequently, we define our parameter space by $\theta = (A, b) \in \mathbb{R}^{p \times m + m}$. For the CartPole-v1 we need 4, for the Swimmer-v1 16 and for the Hopper-v1 36 dimensions. For all environments, we normalize the reward axis. For the Hopper environment, we additionally subtract the survival bonus and use state normalization; find further details in Appendix A.5. We hand-tuned the hyperparameter of GIBO within a reasonable degree, where the hyperparameter for ARS are taken from [1]. In the following, we use the reward over function evaluations (calls of RL environment) as evaluation metric for sample efficiency. We averaged the reported policy rewards over ten trials. In Fig. 6 we observe that GIBO reaches the reward thresholds faster and with significantly less variance than ARS.

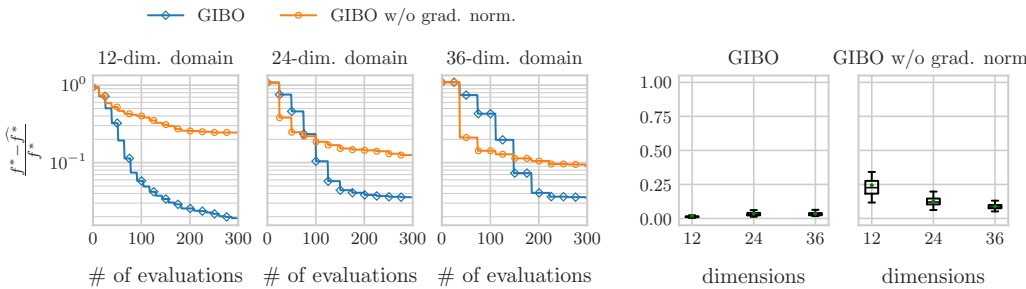

Figure 7: **Ablation study for within-model comparison.** GIBO with and without gradient normalization. Left: Regret over 300 function evaluations. Right: Distribution of regret after 300 function evaluations.

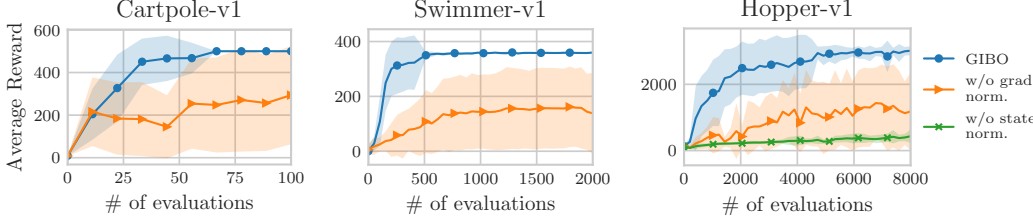

Figure 8: **Ablation study for RL environments.** Training curves of GIBO and its ablated variants on different RL environments, averaged over 10 trials.

## 4.4 Ablation study

In this section, we investigate different implementation choices of the GIBO algorithm. We conduct our ablation experiments on the within-model comparison with synthetic objective functions as well as on RL environments. Gradient normalization using the known GP lengthscales leads to a significant improve in mean performance as well as reduced variance, see Fig. 7. When optimizing policies for the RL benchmarks the GP lengthscales are not known and are learned during training. Fig. 8 shows that even in the case of learned hyperparameters gradient normalization proves to be important for the performance. We applied state normalization only to the Hopper environment and found that for this task it is not possible to learn well-performing policies without this extension. This shows that the normalization of an unknown policy space can be crucial for GP regression.

## 5 Conclusion

We introduce GIBO, a gradient-based optimization algorithm with a BO-type active sampling strategy to improve gradient estimates for black-box optimization problems. When the model assumptions of BO are satisfied, we show that the algorithm is significantly more sample-efficient, especially in higher dimensions, compared to baseline algorithms for black-box optimization.

Additionally, we show the benefits of active sampling and probabilistic gradient estimates with GIBO by solving popular RL benchmarks for which the model assumptions do not hold exactly. When compared to random sampling, GIBO is more sample efficient and has lower variance. Yet, the performance benefits are less pronounced in the RL task. This highlights that GIBO especially shines when prior knowledge is available while it still performs reasonably otherwise. Nonetheless, we want to remark that the prior biases the gradient estimates and wrong assumptions about the objective function can deteriorate performance. However, in some sense, all hyperparameters in RL algorithms encode some form of prior knowledge about the problem at hand. In our view, explicit probabilistic priors are an appropriate and intuitive form of prior knowledge to obtain, e.g., from domain knowledge or available data from prior experiments.

Since it is straightforward to include derivative observations into GIBO, we expect similar improvements for other existing RL methods when integrating our method as an additional layer between gradient estimators and optimizers. The proposed framework can suggest different exploration policies and combine all available data into a posterior belief over the Jacobian. For future research, we want to utilize GIBO with state-of-the-art actor-critic algorithms to improve sample complexity of these methods.

In a more general context, our active sampling methodology makes a step towards autonomous decision-making. GIBO decides on a learning experiment for the autonomous agent. Whenever a decision process is automated, the responsibility for legal and ethical consequences of these decisions must be resolved. However, we do not discuss how the decision-maker, GIBO, can be constrained to ensure compliance with regulatory requirements, which is a relevant aspect for future research.

## Acknowledgments and Disclosure of Funding

The authors thank D. Baumann, P. Berens, A. R. Geist, H. Heidrich and F. Solowjow for their helpful comments and discussions. This work was supported in part by the Cyber Valley Initiative; the Max Planck Society; by the German Federal Ministry of Education Research (BMBF): Tübingen AI Center, FKZ: 01IS18039A; and by the Deutsche Forschungsgemeinschaft (DFG, German Research Foundation) under Germany's Excellence Strategy – EXC number 2064/1 – Project number 390727645. The authors thank the International Max Planck Research School for Intelligent Systems for supporting A. von Rohr and S. Müller.

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
