# A Supplementary material to *Local policy search with Bayesian optimization*

This document contains the Appendix for the paper *Local policy search with Bayesian optimization*. Here, we describe further details about the proposed method and experimental setups to improve the reproducibility of our results. Additionally, a Python implementation as well as scripts to reproduce the presented empirical results presented in Sec. 4 are available at `https://github.com/sarmueller/GIBO`. This Appendix is broken up into several sections

- **A.1 Derivatives of the squared exponential kernel.** First and second derivatives of the squared exponential kernel with respect to the data points.
- **A.2 Derivation of the acquisition function.** A detailed derivation of a simpler form for optimizing our acquisition function.
- **A.3 Gradient normalization.** Background information and intuitive explanation for our algorithmic extension 'gradient normalization'.
- **A.4 Synthetic experiments.** Further information about the synthetic experiments. First, we explain how we find the global optimum of the test functions for within- and out-of-model comparison; second, we present the lengthscale distribution; third, error bars for the within-model experiments are shown; fourth, we show our results for out-of-model experiments.
- **A.5 Gym and MuJoCo.** Details for the Gym and MuJoCo experiments.
- **A.6 Linear quadratic regulator.** Details about the linear quadratic regulator experiment.
- **A.7 Hyperparameters.** Tables with hyperparameters for all experiments.
- **A.8 Software licenses.** Some remarks about our implementation.

## A.1 Derivatives of the squared exponential kernel

The SE kernel is given as

$$k(x_1, x_2) = \sigma_f^2 \exp\left(-\frac{1}{2}(x_1 - x_2)^T L(x_1 - x_2)\right)$$

where the lengthscale matrix $L \in \mathbb{R}^{d \times d}$ could be any positive-semidefinite matrix, but in practice it is often chosen to be a diagonal one $L = \mathrm{diag}(1/\ell_1^2, \ldots, 1/\ell_d^2)$. The derivative of the kernel with respect to the first argument is given by

$$\frac{\partial k(x_1, x_2)}{\partial x_1} = -L(x_1 - x_2)k(x_1, x_2).$$

The derivative of the SE-kernel with respect to the second argument is given by

$$\frac{\partial k(x_1, x_2)}{\partial x_2} = -\frac{\partial k(x_1, x_2)}{\partial x_1} = L(x_1 - x_2)k(x_1, x_2).$$

For the second derivative we get

$$\frac{\partial^2 k(x_1, x_2)}{\partial x_1 \partial x_2} = L\left(I - (x_1 - x_2)(x_1 - x_2)^T L\right) k(x_1, x_2)$$

with the relationship

$$\frac{\partial^2 k(x_1, x_2)}{\partial x_1^2} = \frac{\partial^2 k(x_1, x_2)}{\partial x_2^2} = -\frac{\partial^2 k(x_1, x_2)}{\partial x_1 \partial x_2} = -\frac{\partial^2 k(x_1, x_2)}{\partial x_2 \partial x_1}.$$

In case of $x_1 = x_2 = x$, the second derivative of the SE kernel yields

$$\frac{\partial^2 k(x, x)}{\partial x^2} = L\sigma_f^2.$$

## A.2 Derivation of the acquisition function

Starting again from (4) the expected utility can then be written as the Lebesgue-Stieltjes integral

$$\alpha_{\mathrm{GI}}(\theta|\theta_t, \mathcal{D}) = \int \mathrm{Tr}\left(\Sigma'(\theta_t|\mathcal{D})\right) - \mathrm{Tr}\left(\Sigma'(\theta_t|\{\mathcal{D}, (\theta, y)\})\right) \mathrm{d}F(\theta)$$

where $F(\theta)$ is the distribution function. When optimizing the acquisition function with respect to the next query parameter $\theta \in \mathbb{R}^d$, constants can be omitted and the integral simplifies to

$$\arg\max_{\theta} \alpha_{\mathrm{GI}}(\theta|\theta_t, \mathcal{D}) = \arg\max_{\theta} \int - \mathrm{Tr}\left(\Sigma'(\theta_t|\{\mathcal{D}, (\theta, y)\})\right) \mathrm{d}F(\theta).$$

This can be reformulated to a Riemann integral

$$\arg\max_{\theta} \alpha_{\mathrm{GI}}(\theta|\theta_t, \mathcal{D}) = \arg\min_{\theta} \int_{\mathbb{R}} \mathrm{Tr}\left(\Sigma'(\theta_t|\{\mathcal{D}, (\theta, y)\})\right) \cdot p(f(\theta) = y|\mathcal{D}) \, \mathrm{d}y.$$

A property of a Gaussian distribution is, that the covariance function is independent of the observed targets $y$ as shown in Equation (3). Hence, the acquisition function can further be simplified to

$$\arg\max_{\theta} \alpha_{\mathrm{GI}}(\theta|\theta_t, \mathcal{D}) = \arg\min_{\theta} \mathrm{Tr}\left(\Sigma'(\theta_t|\{\mathcal{D}, (\theta, y)\})\right) \underbrace{\int_{\mathbb{R}} p(f(\theta) = y|\mathcal{D}) \, \mathrm{d}y}_{=1}$$

$$= \arg\min_{\theta} \mathrm{Tr}\left(\Sigma'(\theta_t|[X, \theta])\right)$$

where the variance only depends on a virtual data set $\hat{X} = [\theta_1, \ldots, \theta_n, \theta] =: [X, \theta]$.

## A.3 Gradient normalization

Fist-order methods, like gradient ascent, use the gradient $g_t$ (first derivative) to update their parameters

$$\theta_{t+1} = \theta_t + \eta \cdot g_t(\theta_t).$$

The gradient vector can be divided into magnitude and direction

$$g_t(\theta_t) = \underbrace{\|g_t(\theta_t)\|_2}_{\text{magnitude}} \cdot \underbrace{\frac{g_t(\theta_t)}{\|g_t(\theta_t)\|_2}}_{\text{direction}}.$$

This leads to the integration of the gradient's magnitude into the steplength, defined by

$$\|\theta_{t+1} - \theta_t\|_2 = \eta \cdot \|g_t(\theta_t)\|_2.$$

The parameter update is dividable into a magnitude- (steplength) and a direction-update, both depending on the gradient

$$\theta_{t+1} = \theta_t + \underbrace{\eta \cdot \|g_t(\theta_t)\|_2}_{\text{magnitude}} \cdot \underbrace{\frac{g_t(\theta_t)}{\|g_t(\theta_t)\|_2}}_{\text{direction}}.$$

We can see that the update step inherits its direction and its magnitude from the gradient respectively. While it is beneficial for an optimizer to follow the gradient's direction, research has discovered several problems when using a scaled version of the gradient's magnitude as steplength [42]: (i) divergent oscillation from the optimum, (ii) loss of gradient at plateaus or saddle points, (iii) getting stuck in local optima. Hence, a striking trend in the development of first-order gradient methods is the adaption of the steplength. Many state-of-the-art methods introduce heuristics to estimate proper steplength like Momentum [43], AdaGrad [44], RMSProp [42] or Adam [30].

All presented methods have in common that they use the gradient's direction, but introduce new ideas to set a proper steplength. For our approach, modeling the objective function with a GP, we gain more knowledge about the error surface than the mentioned state-of-the-art methods. More precisely, the hyperparameters of the GP give valuable insights we want to exploit for the steplength of our gradient descent optimization.

One interesting property is that lengthscales of a SE-kernel and correlation length are directly related. For a SE-kernel with outputscale $\sigma_f = 1$ and the same lengthscale $\ell = \ell_1, \ldots, \ell_d$ for every dimension the kernel equation results in

$$k(x, \hat{x}) = \exp\left(-\frac{\|x - \hat{x}\|_2^2}{2\ell^2}\right).$$

For $f \sim \mathcal{GP}(0, k)$ the correlation between $f(x)$ and $f(\hat{x})$ is exactly $k(x, \hat{x})$. With a SE-kernel any two points have positive correlation, but it decreases to zero quickly with increasing distance:

- $\|x - \hat{x}\|_2 = \ell$, the correlation is $\exp(-\frac{\ell^2}{2\ell^2}) = \exp(-\frac{1}{2}) \approx 0.61$,
- $\|x - \hat{x}\|_2 = 2\ell$, the correlation is $\exp(-\frac{2^2}{2}) \approx 0.14$,
- $\|x - \hat{x}\|_2 = 3\ell$, the correlation is $\exp(-\frac{3^2}{2}) \approx 0.01$.

Because of the equivalence of lengthscales and correlation length for the SE-kernel, it appears natural to set the steplength proportional to the lengthscales. Therefore, we normalize the gradient using the SE lengthscales $L$

$$\hat{g}_t = \mathbb{E}\left[\nabla_\theta J(\theta)\right]\big|_{\theta = \theta_t}, \ \Delta\theta_t = \frac{\hat{g}_t}{\|\hat{g}_t\|_L},$$

where $\|x\|_L = \sqrt{x^T L x}$ is the Mahalanobis norm. We update the parameters with

$$\theta_{t+1} = \theta_t + \eta \cdot \Delta\theta_t.$$

With this extension, the constant stepsize $\eta$ is the proportional factor for scaling the lengthscales for the steplength. For instance a stepsize of $\eta = 1$ means our steplengths are the lengthscales for every search direction, resulting in a correlation of approximately $0.61$ between our new parameters $\theta_{t+1}$ and our old parameters $\theta_t$. This leads to a much more intuitive way to set a stepsize. Moreover, with a hyperparameter optimization for our GP model we adapt not only the lengthscales but also the steplengths for every search direction.

### A.4 Synthetic experiments

To calculate the regret, the global optimum of each test function was approximated by local optimization with a much higher sample budget. The start point of the local optimization was the best point of the 1000 sampled function values. This information was never revealed to the algorithms under test. After each parameter update, the algorithms were asked to return the best-sampled point in the input space so far, which yields the regret curves in Fig. 3 and Fig. 11.

**Lengthscale distribution**

To be able to perform similar computationally expensive experiments with the same number of training samples in higher dimensional domains, lengthscales were scaled with the expected distance $\Delta(d)$ between randomly picked points from a unit $d$-dimensional hypercube. There is no closed form solution for this hypercube line picking, but it can be bounded with [45]

$$\frac{1}{3}d^{1/2} \leq \Delta(d) \leq \left(\frac{1}{6}d\right)^{1/2}\sqrt{\frac{1}{3} + \left[1 + 2\left(1 - \frac{3}{5d}\right)^{1/2}\right]}.$$

The upper and lower bound are shown in blue and orange, respectively, in Fig. 9. To be still comparable to the experiments from Hennig and Schuler [34], the upper bound is scaled down such that it fulfills $\Delta(2) = 0.1$ for the 2-dimensional domain. The resulting scaled upper bound in green in Fig. 9 serves for an orientation for the chosen lengthscale sample distribution

$$\ell(d) \sim \mathcal{U}(2 \cdot \Delta(d)_{\text{sub}}(1 - \gamma), 2 \cdot \Delta(d)_{\text{sub}}(1 + \gamma)),$$

in red in Fig. 9, where $\Delta(d)_{\text{sub}}$ is the scaled upper bound function and $\gamma = 0.3$ corresponds to the noise parameter.

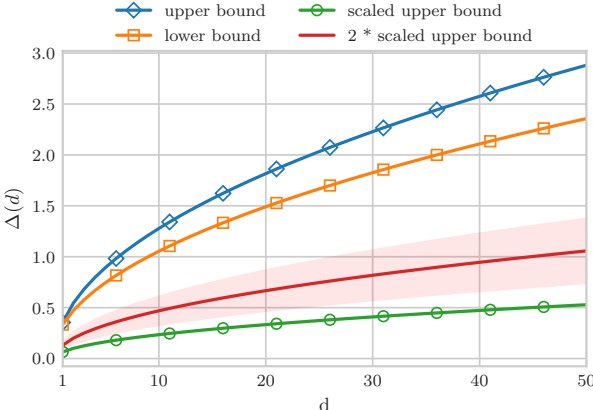

Figure 9: Lengthscale sample distribution: The image shows approximations of the expected distance $\Delta(d)$ between two randomly picked points in a unit domain.

## Within-model comparison

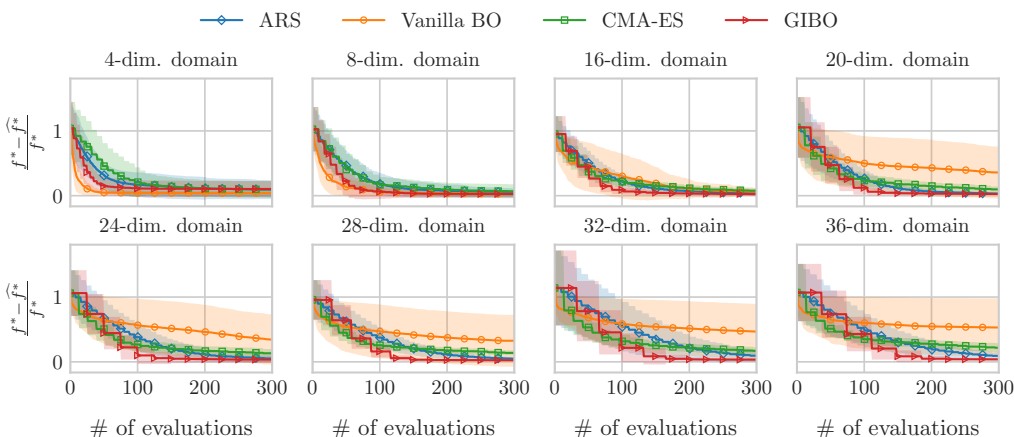

Figure 10: Standard deviation of within-model comparison: We show the same results as in Fig. 3, but include the standard deviation (shaded region) over the 40 objective functions per domain and therefore show the regret on a linear scale.

As with Fig. 4, the error bars in Fig. 10 show consistently lower variance in regret of GIBO compared to the benchmark algorithms.

### Out-of-model comparison

For the out-of-model comparison, we sample the objective function from the same prior as in the within-model comparison 4.1. However, the true parameters of the prior distribution are not revealed to the GP-based algorithms to investigate the effect of model mismatch and hyperparameter optimization. We set proper hyperprior distributions for GIBO and Vanilla BO to perform maximum a posteriori (MAP) estimation for hyperparameters determination from data. The noise of the likelihood is fixed to the true value $\sigma_n = 0.1$, since this value can usually be estimated easily in additional experiments. Since the GP-based methods had to learn their hyperparameters, we no longer scaled the hyperparameters of ARS and CMA-ES with the mean of the lengthscale's sample distribution.

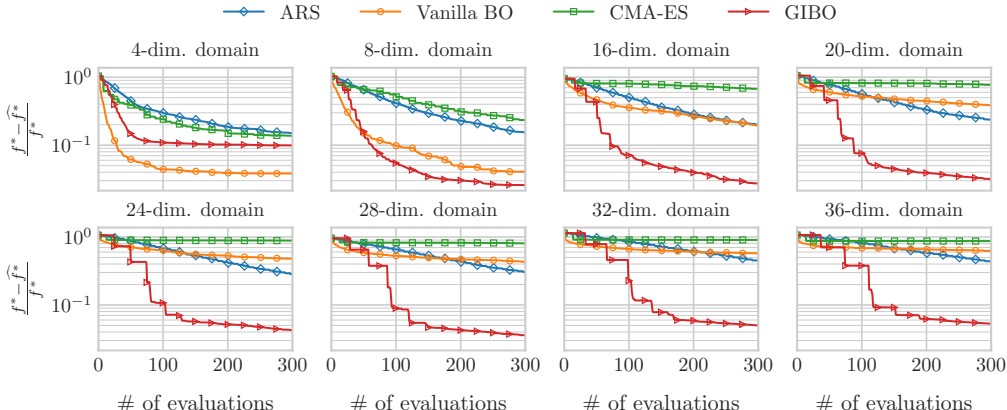

Figure 11: Out-of-model comparison: Mean of the normalized distance of the function value at optimizers' best guesses from the true global maximum for eight different dimensional function domains. For the GP-based methods, hyperparameters were optimized. 40 runs. Logarithmic scale.

Fig. 11 shows similar performance of the GP based methods for the within- and out-of-model comparison. This can be interpreted as a result of a well performing hyperparameter optimization, when proper hyperpriors are given. The most obvious difference is the performance change of ARS and CMA-ES. With no scaling of the space-dependent hyperparameters of these algorithms, i.e., prior knowledge of the objective function, the performance decreases drastically compared to GIBO. We interpret these results such that GIBO is able to learn relevant properties of the objective function, using the available data points and the hyperpriors effectively. This shows the benefits of the probabilistic model of the objective function even when hyperparameter are not known exactly.

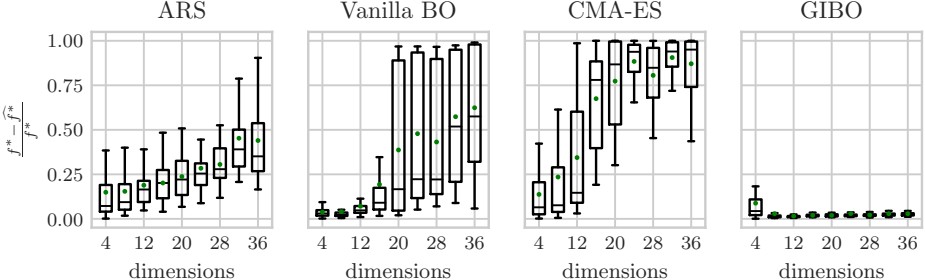

Figure 12: Out-of-model comparison: Boxplots (40 runs) show the the normalized distance of optimizers' best found values after 300 function evaluations from true global maximum. For the GP-based methods, hyperparameters were optimized. The whiskers lengths are 1.5 of the interquartile range; the black horizontal lines represent medians, green dots the means.

In Fig. 12 and Fig. 13 we can see that only our proposed algorithm seems to be able to maintain performance, despite the need for hyperparameter optimization. This can be explained by only having a local model of the function, which results in an easier hyperparameter optimization.

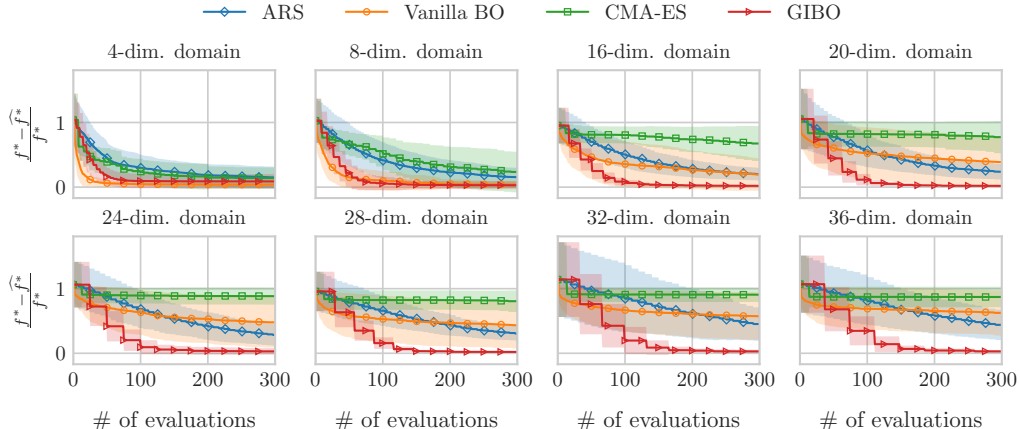

Figure 13: Standard deviation of within-model comparison: We show the same results as in Fig. 11, but include the standard deviation (shaded region) over the 40 objective functions per domain and therefore show the regret on a linear scale.

## A.5 Gym and MuJoCo

**CartPole-v1.** The linear policy for CartPole maps 4 states to 2 discrete actions. With the help of a case distinction

$$\pi_\theta(s) = \begin{cases} 1 & As > 0 \\ 0 & \text{else} \end{cases}$$

this is realized with only 4 parameters, integrated in $A \in \mathbb{R}^4$. During training we normalized the reward axis for GIBO with the maximum achievable reward $r_t = r_t/500$, making it easier to model a GP to the policy space.

**Swimmer-v1.** The linear policy for Swimmer $\pi_\theta$ consists of 16 parameters, for $A \in \mathbb{R}^{8 \times 2}$. We again normalized the reward axis with $r_{\max} = r_t/350$.

**Hopper-v1.** The Hopper MuJoCo locomotion tasks needs a search space of 36 dimensions, integrated into an affine linear policy with $A \in \mathbb{R}^{11 \times 3}$ and $b \in \mathbb{R}^3$. In the work of Mania et. al [1] they showed an increase in performance for the Hopper environment when making use of the state normalization. Therefore, both algorithms are using this algorithmic extension. Moreover, the reward is manipulated by subtracting the survival bonus and normalizing it $r_t = (r_t - 1)/1000$.

## A.6 Linear quadratic regulator

For the LQR experiment a discrete time infinite horizon average cost LQR problem with additive i.i.d. Gaussian noise is considered and can be formalized with

$$\min_{u_0, u_1, \dots} \lim_{T \to \infty} \frac{1}{T} \mathbb{E} \left[ \sum_{t=0}^{T-1} x_t^T Q x_t + u_t^T R u_t \right]$$

$$\text{s.t. } x_{t+1} = A x_t + B u_t + w_t.$$

With discrete-time index $t \in \mathbb{N}$, state $x_t \in \mathbb{R}^n$, control input $u_t \in \mathbb{R}^p$, system matrix $A \in \mathbb{R}^{n \times n}$, $B \in \mathbb{R}^{n \times p}$, $Q \in \mathbb{R}^{n \times n}$, $R \in \mathbb{R}^{p \times p}$, and the independent identically distributed (i.i.d.) Gaussian noise $w_t \sim N(0, W)$. The system is assumed to be $(A, B)$-stabilizable. Hence, the optimal control law is a stationary linear feedback policy $u_t = K x_t$ and the feedback gain $K \in \mathbb{R}^{p \times n}$ is given by solving the discrete algebraic Ricatti equation

$$P = A^T P A - A^T P B (R + B^T P B)^{-1} B^T P A + Q,$$

setting

$$K = -(R + B^T P B)^{-1} B^T P A.$$

We consider the LQR instance from [1] (also used in [40], originally from [41]), a challenging instance for LQR with unknown dynamics and

$$A = \begin{bmatrix} 1.01 & 0.01 & 0 \\ 0.01 & 1.01 & 0.01 \\ 0 & 0.01 & 1.01 \end{bmatrix}, \ B = I, \ Q = 10^{-3}I, \ R = I$$

with $n = 3$ and $p = 3$. The matrix $A$ has eigenvalues greater than 1, hence the system is unstable without control. Moreover, with a control signal of zero the system has a spectral radius of $\rho \approx 1.024$ resulting in slowly diverging states. Hence, long trajectories are required to evaluate the performance of the controller.

Our metric of interest is the relative error $\frac{J(\hat{K}) - J_*}{J_*}$, where $J_*$ is the optimal infinite horizon cost on the average cost objective, and $J(\hat{K})$ is the infinite horizon cost of using the controller $\hat{K}$ in feedback with the true system specified. The exact calculation of the metric is given for $\hat{K}$ that stabilizes $(A, B)$ in Lemma 4.0.5 of the technical report [46] with

$$J(\hat{K}) - J_* = \text{Tr}\,(W\hat{P}) - \text{Tr}\,(WP)$$
$$= \text{Tr}\,(\Sigma(\hat{K})(\hat{K} - K)^T(R + B^T PB)(\hat{K} - K))$$

where Tr the trace operator and $\Sigma(\hat{K})$ the stationary covariance matrix of $(A, B)$ in feedback with $\hat{K}$. $\Sigma(\hat{K})$ is solvable with the discrete Lyapunov equation

$$\Sigma(\hat{K}) = (A + B\hat{K})\Sigma(\hat{K})(A + B\hat{K})^T + W.$$

The experiments were run by collecting $M$ independent trajectories of length $N = 300$ of the system specified above. This produces a collection of $MN$ tuples $D = \left\{ \left( x_k^{(l)}, u_k^{(l)}, r_k^{(l)}, x_{k+1}^{(l)} \right) \right\}_{k=1, l=1}^{N,M}$. The process is repeated 100 times. In our experiments we will refer to the value $M \cdot N$ as the number of timesteps, and each set $D$ of $MN$ tuples as a trial. The optimized reward is defined by the negative quadratic cost of the LQR problem. Since the cost is blowing up when the controller is unstable, the reward is manipulated to

$$r_k^{(l)} = -\log(1 - r_k^{(l)}).$$

## A.7 Hyperparameters

**Synthetic experiments**

Table 1: Hyperparameters (and hyperpriors) for the synthetic within-model and out-of-model experiments. $d$ refers to the dimension of the domain. $\ell(d)$ is the lengthscale's sample distribution and $2 \cdot \Delta(d)_{\text{sub}}$ its mean. The operator $//$ refers to integer floor division. VBO stands for 'Vanilla BO'.

| Method | Hyperparameters | Within-model | Out-of-model |
|---|---|---|---|
| ARS | $\alpha$ | 0.02 | 0.02 |
| | $\nu$ | $0.1 \cdot 2 \cdot \Delta(d)_{\text{sub}}$ | 0.01 |
| | $N$ | $1 + d//8$ | $1 + d//8$ |
| CMA-ES | $\sigma$ | $0.3 \cdot \Delta(d)_{\text{sub}}$ | 0.5 |
| GIBO & VBO | lengthscales | $\ell(d)$ | $\ell(d)$ |
| | signal variance $\sigma_f$ | 1.0 | $\mathcal{U}(0.1, 5)$ |
| | likelihood noise $\sigma_n$ | 0.1 | 0.1 |
| GIBO | optimizer | SGD | SGD |
| | $\eta$ | 0.25 | 0.25 |
| | $M$ | $d$ | $d$ |
| | $N_m$ | $5 \cdot d$ | $5 \cdot d$ |
| | $\delta_b$ | 0.2 | 0.2 |
| | norm. gradient | True | True |

**Linear quadratic regulator**

Table 2: Hyperparameters (and hyperpriors) for the LQR experiment.

| Method | Hyperparameters | LQR |
|---|---|---|
| | lengthscales | $\mathcal{U}(0.01, 0.3)$ |
| | signal variance $\sigma_f$ | $\mathcal{N}(20, 5)$ |
| | likelihood noise $\sigma_n$ | 2 |
| | optimizer | SGD |
| GIBO | $\eta$ | 1. |
| | $M$ | 9 |
| | $N_m$ | 40 |
| | $\delta_b$ | 0.1 |
| | norm. gradient | True |

**Gym and MuJoCo**

Classic control and MuJoCo (mujoco-py v0.5.7) tasks included in the OpenAI Gym-v0.9.3.

Table 3: Hyperparameters (and hyperpriors) for Gym and MuJoCo experiments.

| Method | Hyperparameters | CartPole-v1 | Swimmer-v1 | Hopper-v1 |
|---|---|---|---|---|
| | $\alpha$ | 0.025 | 0.02 | 0.01 |
| ARS | $\nu$ | 0.02 | 0.01 | 0.025 |
| | $N$ | 8 | 1 | 8 |
| | $b$ | 4 | - | 4 |
| | lengthscales | $\mathcal{U}(0.01, 0.3)$ | $\mathcal{U}(0.01, 0.3)$ | $\mathcal{U}(0.01, 0.5)$ |
| | signal variance $\sigma_f$ | $\mathcal{N}(2, 1)$ | $\mathcal{N}(2, 1)$ | $\mathcal{N}(2, 1)$ |
| | likelihood noise $\sigma_n$ | 0.5 | 0.01 | 0.01 |
| | optimizer | SGD | SGD | SGD |
| | $\eta$ | 1. | 0.5 | 0.5 |
| GIBO | $M$ | 8 | 16 | 8 |
| | $N_m$ | 20 | 32 | 48 |
| | $\delta_b$ | 0.1 | 0.1 | 0.2 |
| | norm. gradient | True | True | True |
| | state norm. | False | False | True |

## A.8 Software licenses

The implemention of GIBO is based on GPyTorch [47] and BoTorch [31] both published under the MIT License.

The RL benchmarks provided by the OpenAI Gym [35] published under the MIT License and the MuJoCo pyhsics engine [36] has a proprietary license `https://www.roboti.us/license.html`.