# OpenReview forum: "Local policy search with Bayesian optimization"
_NeurIPS.cc/2021/Conference — NeurIPS 2021 Poster_

### Official Review · Reviewer_h5kV · 2021-07-04

**Rating:** 7
**Confidence:** 4

**Summary:**

This paper modifies the standard BayesOpt pipeline in 2 ways:

1. Defines a new acquisition function, the "Gradient Information" (GI) which is the reduction in the variance of the Jacobian variance of $J$.
2. Exploration and Exploitation become disjointed: During the BayesOpt process, a batch of $M$ queries are made via argmax of the GI acquisition, purely for exploration (rather than a mix of exploitation and exploration as is standard in BayesOpt), with the goal of producing a final gradient update step $\nabla_{\theta}J$.

This defines a "GIBO" algorithm, which is then experimentally evaluated over the following benchmarks:

1. Sampled functions from GP's
2. LQR Controllers
3. Linear controllers for standard Mujoco tasks.

which show that GIBO outperforms previous ES/ARS and standard BayesOpt methods especially in high dimensional settings, most likely due to the use of "local search"/gradients.

**Limitations And Societal Impact:**

Yes

**Main Review:**

Pros:
1. I overall think this is a promising and good paper, with many potential future directions in BayesOpt + ES/RL research, especially due to its strengths on high dimensional optimization.
2. In some sense, the method can be seen as a way of "intelligently" picking a batch of neighborhood points to form a gradient, rather than e.g. simply via sampling perturbations from an Isotropic Gaussian (as is standard in ARS/ES methods).
3. Paper is well-written, and well-understandable in general for someone familiar with BayesOpt methods.

Questions/Cons:
1. I'm assuming that because the acquisition function is based on the variance of the Jacobian at the "current" point $\theta_{t}$, this means that future argmax query points in the batch would likely be very close to $\theta_{t}$ right? (Since local information around $\theta_{t}$ probably contributes more to the gradient estimation at $\theta_{t}$). I believe that this could potentially cause problems with global exploration / have the same issues with gradient estimation methods converging too fast to local minima.
2. Citation requests: You may want to cite/compare e.g. [1] (which uses a k-DPP to maintain unbiased gradient estimation in ES, but reduce variance and improve exploration) and other related "manual" methods on improving the variance of ES gradient estimation. Similar to the previous point as well: how "exploratory" are the queries really, when optimizing the GI acquisition?
3. How well does GIBO perform in the basic neural network policy setting (where there will be ~200-1000 parameters from a e.g. 2-layer MLP)? Such scenarios are still quite common even in ES + RL methods (e.g. A nonlinear network is needed for training on Pendulum with ES from my experience; a linear policy doesn't always work)
4. How parallelizable is the method, especially with the $M$ batch query section? A major benefit of ARS/ES is that they can be parallelized with an arbitrary number of workers, (analogous to "$M$"), but it seems like the dataset update $\mathcal{D} \leftarrow  (\theta,y) $ forces sequentiality. Could this be moved outside of the batch-for-loop?

[1] Structured Monte Carlo Sampling for Nonisotropic Distributions via Determinantal Point Processes

Small notation issues:
* In Algorithm 1, $m$ is already used as a variable denoting the mean function before; perhaps change it for clarity.

**Time Spent Reviewing:**

4

---

> ### Author Response · Authors · 2021-08-10
> **Reply to reviewer h5kV**
>
> Thank you for your thoughtful review and detailed questions.
>
> > I'm assuming that because the acquisition function is based on the variance of the Jacobian at the "current" point , this means that future argmax query points in the batch would likely be very close to  right? (Since local information around probably contributes more to the gradient estimation at ). I believe that this could potentially cause problems with global exploration / have the same issues with gradient estimation methods converging too fast to local minima.
>
> Correct, GIBO is a local optimizer that comes with the possibility to converge to local minima. We can see this behavior in the low dimensional synthetic test cases where global optimization is able to outperform both local methods easily. However, as we argue in the paper, for high dimensional search spaces, a global optimum is oftentimes not achievable in sample efficient manner.
>
> > Citation requests: You may want to cite/compare e.g. [1] (which uses a k-DPP to maintain unbiased gradient estimation in ES, but reduce variance and improve exploration) and other related "manual" methods on improving the variance of ES gradient estimation. Similar to the previous point as well: how "exploratory" are the queries really, when optimizing the GI acquisition?
>
> We will look more carefully at the related literature on ES. Thank you for the pointers.
>
> The explicit goal of the exploration in GIBO is to reduce the uncertainty of the gradient *at the current point*. Therefore the exploration will be local in nature, similar to how a gradient-based optimizer *explores* locally with each update step. By design, no global exploration takes place.
>
> > How well does GIBO perform in the basic neural network policy setting (where there will be ~200-1000 parameters from a e.g. 2-layer MLP)? Such scenarios are still quite common even in ES + RL methods (e.g. A nonlinear network is needed for training on Pendulum with ES from my experience; a linear policy doesn't always work)
>
> We have not done any experiments learning the weights of ANNs. We are planning these for future work. We have chosen to stick to linear policies since they are sufficient to learn policies on MuJoCo benchmarks, as has been shown in [1].
>
> > How parallelizable is the method, especially with the $M$ batch query section? A major benefit of ARS/ES is that they can be parallelized with an arbitrary number of workers, (analogous to "$M$"), but it seems like the dataset update $\mathcal{D} \leftarrow (\theta, y)$ forces sequentiality. Could this be moved outside of the batch-for-loop?
>
> In principle, one could easily parallelize the inner loop (line 7 of the algorithm) when we solve equation (7) for a batch of points instead of for a single point. This would mean we find the $M$ query points that minimize the uncertainty about the gradient. Such approach could be promising and more efficient than sequential sampling and will be part of our future research.
>
> Another way to parallelize the algorithm is by simply starting GIBO in different policy space regions, which would correspond to local multi-start optimization.
>
> However, as is the case with gradient-based optimizers, the outer loop cannot be parallelized easily.
>
>
> [1] Horia Mania, Aurelia Guy, and Benjamin Recht. Simple random search of static linear policies is competitive for reinforcement learning. In Advances in Neural Information Processing, Systems 31, pages 1800–1809. 2018.

---

### Official Review · Reviewer_b2oQ · 2021-07-06

**Rating:** 6
**Confidence:** 4

**Summary:**

In this paper, the authors propose a new acquisition function for Bayesian optimization based on gradient information. A GP model estimates the Jacobian and its estimation is used for gradient update. This method is implemented in the setting of policy search for reinforcement learning. Empirical evaluations are provided for a synthetic setting, LQR and MuJoCo gym environments.

**Limitations And Societal Impact:**

The potential scalability limitation needs more discussion.

**Main Review:**

Post-rebuttal comments: thanks for providing a detailed response. I still think the comparison with [1] is valuable. Or as the authors mentioned, demonstrate how GIBO can be used in conjunction with [1]. As a result, I will keep my score.

Originality: I really like the perspective the authors have taken to connect policy search with Bayesian optimization. The acquisition function based on uncertainty reduction on gradient is novel and fits the use case quite well.

The related work section is comprehensive. I would like to draw the authors’ attention to the following paper [1] Learning Search Space Partition for Black-box Optimization using Monte Carlo Tree Search, Linnan Wang, Rodrigo Fonseca, Yuandong Tian, NeurIPS 2020, which also considers a blackbox setting for policy search.

Quality: The paper is technically sound. I have some technical questions on the algorithm.
1. Can the authors provide more details on how the optimization for equation (7) is done? The appendix mentioned Cholesky factorization to model K(\hat{X}, \hat{X}) but it is unclear to me how the whole equation is optimized. Were you able to compute the gradient analytically and perform gradient descent?

2. Notation-wise, should the \theta_t in equation (7) be \theta_n instead since \hat{X} indicates that n different thetas have been queries?

3. In section 3.3 for the uncertainty threshold, the Sigma’s are matrices, what does it mean to bound the difference between two matrices with a scalar \epsilon?

4. On the hyperparameters in section 3.3, e.g., the bound (\delta) and the threshold (\epsilon), how sensitive is the performance of the GIBO algorithm to them?

5. Can you include error bars for the experiment in section 4.1?

Clarity:This paper is well-written and easy to read. Good job!

Significance: I enjoy the novelty of this paper. However, I think the significance can be improved by improving the empirical section. Comparison with [1] should be included as it is competitive with ARS on Swimmer and Hopper for MuJoCo.

Moreover, I would like to know if it is possible to consider higher dimensional environments, such as the humanoid. Is there a potential scalability issue with the proposed method?

**Time Spent Reviewing:**

2

---

> ### Author Response · Authors · 2021-08-10
> **Reply to reviewer b2oQ**
>
> Thank you for your thoughtful review and detailed questions.
>
> > Can the authors provide more details on how the optimization for equation (7) is done? The appendix mentioned Cholesky factorization to model K(\hat{X}, \hat{X}) but it is unclear to me how the whole equation is optimized. Were you able to compute the gradient analytically and perform gradient descent?
>
> Equation (7) is optimized using a local and multi-start black-box optimization scheme supplied by BoTorch (L-BFGS-B). So no gradients are used.
>
> The Cholesky factor update is computationally cheaper than recalculating the factorization from scratch, which would be $O(N^3)$.
>
> > Notation-wise, should the \theta_t in equation (7) be \theta_n instead since \hat{X} indicates that n different thetas have been queries?
>
> The notation is correct, the term $K(\theta_t, \hat{X})$ represents the kernel matrix for the current parameter $\theta_t$ and the extended data set $\hat{X} = [\theta_1, \dots, \theta_n, \theta] \in \mathbb{R}^{d\times n+1}$. It results from the derivation of the acquisition function in Appendix A.2 and corresponds to one term of the minimization of the Jacobian's variance at $\theta_t$. The equation for the Jacobian's variance is given with equation (3). On a more intuitive level, we want to reduce the Jacobian's variance **at $\theta_t$** when a new point $\theta$ is sampled.
>
>
> > In section 3.3 for the uncertainty threshold, the Sigma's are matrices, what does it mean to bound the difference between two matrices with a scalar \epsilon?
>
> This is a mistake in the paper. We compare the traces of the two matrices.
>
> > On the hyperparameters in section 3.3, e.g., the bound (\delta) and the threshold (\epsilon), how sensitive is the performance of the GIBO algorithm to them?
>
>  The bound for the acquisition function $\delta_b$ is a parameter for computational convenience. If it is not *too small* w.r.t. to the lengthscales, the algorithm is not sensitive to this parameter.
>
> For the uncertainty threshold $\epsilon_a$, the algorithm can make update steps earlier, which slightly improves sample efficiency. Again if a reasonably small value w.r.t. to the signal variance of the GP is chosen, the algorithm is not very sensitive to this parameter.
>
> Since we are planning an ablation study, the influences of these extensions and parameters will be added to the paper.
>
> > Can you include error bars for the experiment in section 4.1?
>
> Yes, but the readability of Figure 3 would greatly suffer, also due to the log scale. As a compromise, we already added the error bars for step 300 in Figure 4.
>
> We can add Figure 3 with error bars in the appendix, but the graph will be hard to read.
>
> >  Comparison with [1] should be included as it is competitive with ARS on Swimmer and Hopper for MuJoCo
>
> We are purposely comparing against a simple baseline (ARS) to highlight the benefits of active sampling for gradient information. Since the GIBO algorithm can be used in conjunction with most policy gradient algorithms, we forgo the comparison against SOA algorithms.
>
> We think the more interesting question is how [1] can be used in conjunction with GIBO by e.g., globally learning multiple promising regions and restarting GIBO in those learned regions.
>
> > Moreover, I would like to know if it is possible to consider higher dimensional environments, such as the Humanoid. Is there a potential scalability issue with the proposed method?
>
> Scalability issues arise from calculating the GP posterior ($\mathcal{O}(N^3)$).
>
> In principle, it is possible to run GIBO on significantly higher dimensional environments since we only have a local GP model of the last $N_m$ samples. However, we still need 'enough' points to model the local environment, which is still feasible for the Humanoid environment.
>
> However, noteworthy is the limitation that our algorithms need 'good enough' prior information in the form of a GP kernel function and hyperpriors for the objective modeling.
> Despite the state normalization for RL environments, this is still a challenge for higher dimensions, especially since we restricted ourselves to a SE kernel.
> Hence, we will leave objective function modeling and empirical evaluation for higher dimensional RL problems such as Humanoid for future work. Here, we also want to work with gradient samples or space partitioning.

---

### Official Review · Reviewer_h27d · 2021-07-16

**Rating:** 5
**Confidence:** 4

**Summary:**

This paper concerns the high sample complexity of policy gradient methods and the computational complexity of Bayesian optimization. The authors propose a novel algorithm called GIBO (and its local variants) to minimize uncertainty in gradient estimation. Experiments are conducted and compared with some naive baselines.

**Limitations And Societal Impact:**

The authors state the limitations of their algorithms that rely on prior knowledge of the objective. Negative social impacts are also listed in the paper.

**Main Review:**

Originality and significance:
The idea itself is interesting. However, the contributions and improvements over GP algorithms are unknown. See detailed comments below.

Clarity:
The paper has clear structures and is easy to follow.

Detailed comments:

1. The authors claim that BO methods have been restricted to low-dimensional problems. But there are modern Bayesian RL algorithms that are both theoretically grounded and verified experimentally in high-dimensional tasks such as Mujoco [1]. It's thus not clear the contributions of this work.

2. The experiments are conducted and compared with only very simple baselines. Since the paper aims to address the drawbacks of policy gradient with random perturbations algorithms and BO algorithms, it will be more convincing to compare with more SOTA methods in these two areas.

3. In Fig. 3, it seems that the curves are not converging in the plots. So the improvements of GIBO are not clear.


[1] Sebastian Curi et al. Efficient Model-Based Reinforcement Learning through Optimistic Policy Search and Planning. NeurIPS 2020.


**Time Spent Reviewing:**

4

---

> ### Author Response · Authors · 2021-08-10
> **Reply to reviewer h27d**
>
> We thank the reviewer for the presented discussion points. Please find our responses below.
>
> > However, the contributions and improvements over GP algorithms are unknown.
>
> We are not quite sure what the reviewers mean when referring to *GP algorithms*. While we have chosen a GP to model uncertainty over the objective, this is just one choice for a probabilistic model. We would not consider GIBO to be inherently linked to GPs.
>
> We will revise the paper to clarify that our contribution is **not** a new state-of-the-art performance on MuJoCo tasks.
> Instead, we have presented an optimization scheme for actively exploring the objective's gradient to be used in a gradient-based optimizer. We have also empirically shown that active exploration leads to an improved sample efficiency compared to random exploration and reduced variance in performance when prior knowledge is available.
>
> For RL tasks where prior knowledge is not readily available, the method still performs at least on par with the baseline.
>
> To our knowledge, this is a novel contribution.
>
>  > The authors claim that BO methods have been restricted to low-dimensional problems. But there are modern Bayesian RL algorithms that are both theoretically grounded and verified experimentally in high-dimensional tasks such as Mujoco [1]. It's thus not clear the contributions of this work.
>
> We are considering a black-box zeroth-order optimization problem in the context of policy search methods. The cited reference is a model-based RL (MBRL) algorithm, which is quite different.
>
> However, we shall extend the related work section with a discussion on Bayesian MBRL.
>
> > In Fig. 3, it seems that the curves are not converging in the plots. So the improvements of GIBO are not clear.
>
> Figure 3 shows improved **sample complexity**. We do not claim improved convergence properties. When converged, the global method with regret guarantees will consistently outperform the local ones. The main problem with BO is that it converges extremely slowly in high-dimensional search spaces and, when used with GPs, it is computationally infeasible to run BO until convergence.
>
> We shall make this point clearer when we discuss the presented results.

---

### Official Review · Reviewer_up9D · 2021-07-16

**Rating:** 6
**Confidence:** 5

**Summary:**

The paper introduces GIBO, a modification of Bayesian Optimization (BO) that performs local policy search. Compared to classic BO, the strengths of this method are that it does not require storing all the past data points, making it more computational efficient, and that it is more stable when moving to higher dimensional problems. The authors compare their method with standard BO and Augmented Random Search, an approach based on finite difference policy search.

**Limitations And Societal Impact:**

Yes

**Main Review:**


I found the paper well written and easy to follow. The authors related their method to previous approaches using BO for Reinforcement Learning. Since the experiments compare GIBO with CMA-ES and ARS, I would have expected to see a discussion about the similarities and differences with other evolutionary approaches.  Another method that is closely related is Parameter-based Value Functions [1], which directly tries to estimate $J(\theta)$ using a deterministic function. They also include a comparison with ARS using linear and nonlinear policies.
Besides the related work section that could be improved, there are some main issues that prevents acceptance:

- Section 3.3 presents many improvements on the basic idea of GIBO, but there are no ablations in the paper supporting these additions. Is also Vanilla BO using these improvements? Ablating these components in GIBO and Vanilla BO seems necessary here to understand why the proposed method can be considered as an improvement

- One main claim in the paper is that it has low variance compare with other methods (ARS). However, in the experimental evaluation, only 3 trails are used for each environment. I would expect to see at least 10 or 20 runs, especially in these relatively small tasks involving very low dimensional linear policies. Using 3 runs is not enough to assess if the variances are really difference.

- I am a bit concerned about the hyperparameters for GIBO in Table 3. How were these chosen? Are the 3 runs in Figure 6 using different seeds with respect to those that found when optimizing the hyperparameters?

If the authors can expand their related work section and solve the main concerns in the experimental section, I am willing to increase the score for this submission.

[1] Francesco Faccio, Louis Kirsch, and Jurgen Schmidhuber. Parameter-based value functions. In ¨ Int. Conf. on Learning Representations (ICLR), Virtual only, May 2021.

------------------------------------------------------------------------------------------------------------------------------------------------------------
Edit post-rebuttal: The authors clarified my concerns

**Time Spent Reviewing:**

5

---

> ### Author Response · Authors · 2021-08-10
> **Reply to reviewer up9D**
>
> Thank you for your thoughtful review and constructive criticism. We agree that the reference on parameter-based value functions should be discussed in the related work, which we will extend. The pointer to the paper is very much appreciated.
>
> > Section 3.3 presents many improvements on the basic idea of GIBO, but there are no ablations in the paper supporting these additions. Is also Vanilla BO using these improvements? Ablating these components in GIBO and Vanilla BO seems necessary here to understand why the proposed method can be considered as an improvement
>
> We agree that an ablation study for the improvements would be valuable. We tested their effectiveness while developing the algorithm and will include a systematic ablation study for the following extensions:
> - Uncertainty threshold
> - Gradient normalization
> - State normalization (for one of the RL problems, since it cannot be used in the synthetic case)
>
> The local approximations do not change the optimizer's behavior in any significant way and are used purely for computational ease. This is a benefit of using local criteria for the acquisition function.
>
> It would, however, not be a problem to show this empirically in an ablation study in the appendix.
>
> The extensions do not apply to Vanilla BO:
> A **local GP** or **local acquisition function optimization** would stop the global exploration of vanilla BO which is exactly what we are comparing our approach against.
> **Uncertainty threshold** and **Gradient normalization** cannot be used since no gradient is computed in vanilla BO.
> **State normalization** can be used in the RL task, which we we do not benchmark with vanilla BO.
>
> > One main claim in the paper is that it has low variance compare with other methods (ARS). However, in the experimental evaluation, only 3 trials are used for each environment. I would expect to see at least 10 or 20 runs, especially in these relatively small tasks involving very low dimensional linear policies. Using 3 runs is not enough to assess if the variances are really difference.
>
> We choose 3 random seeds to compare our results to the results published in the ARS paper [1].
> We have now run 10 runs for each of the MuJoCo tasks, and the preliminary results show the difference of both algorithms is less pronounced. We will conduct 20 runs in total, update the paper accordingly, and adjust the claims made to the results.
>
> > I am a bit concerned about the hyperparameters for GIBO in Table 3. How were these chosen? Are the 3 runs in Figure 6 using different seeds with respect to those that found when optimizing the hyperparameters?
>
> We adopted the hyperparameters chosen in the ARS paper [1] to meaningful equivalences in our algorithm. The other hyperparameters are tuned, approximately by hand, on different random seeds than the presented results.
>
> [1] Horia Mania, Aurelia Guy, and Benjamin Recht. Simple random search of static linear policies is competitive for reinforcement learning. In Advances in Neural Information Processing, Systems 31, pages 1800–1809. 2018.

---

> > ### Author Response · Authors · 2021-08-19
> > **Results of the ablation study**
> >
> > We have finished the ablation study for the RL tasks as well as the synthetic functions in higher dimensions.  We found that, as expected, the gradient normalization is crucial for the performance and variance of the algorithm.
> > In the synthetic example in 12 dimensions the normalized regret after 300 evaluations improves from $\sim 25^-2$ to $\sim 20^3$ with gradient normalization.  For 24 dimensional synthetic functions regret improves from $\sim 15^-2$ to $\sim 40^3$ and in the 36 dimensional from $\sim 90^-3$ to $\sim 40^3$.  The RL tasks cannot be solved without gradient normalization, leading to very low rewards.
> >
> > For the 'uncertainty threshold' extension the results show only little improvement for the synthetic benchmarks.
> > For the RL tasks the GIBO algorithm performs significantly better, both in mean performance and variance, without the uncertainty threshold. Averaged over 10 random seeds GIBO achieves higher performance with less variance for the Swimmer and similar performance with less variance for Hopper compared to ARS when turning off the 'uncertainty threshold'.
> >
> > Our hypothesis is that, since we tuned $\epsilon_a$ on a different optimization task and by hand, the chosen value was too high for the given tasks.  Since it seems tuning $\epsilon_a$ has a more significant impact on the performance than we initially observed we will exclude this extension from the final paper, thereby simplifying the algorithm and hyperparameter tuning.

---

> > > ### Comment · Reviewer_up9D · 2021-08-25
> > > **The authors clarified my concerns**
> > >
> > > I thank the authors for clarifying my concerns. The ablation performed shows the importance of the main components in the method and I am more convinced about the reduction in variance. I will update my score to marginally acceptance.

---

### Decision · Program_Chairs · 2021-09-27

**Decision:**

Accept (Poster)

**Comment:**

The reviewers appreciate the novelty of the main contribution of the paper, ie the acquisition function defined as reduction in gradient variance under a GP model. The paper could be strengthened further by applying the proposed method to higher-dimensional problems.
Nevertheless, the paper is acceptance.